# Chemical Constituents of the Deep-Sea-Derived *Penicillium solitum*

**DOI:** 10.3390/md19100580

**Published:** 2021-10-17

**Authors:** Zhi-Hui He, Jia Wu, Lin Xu, Man-Yi Hu, Ming-Ming Xie, You-Jia Hao, Shu-Jin Li, Zong-Ze Shao, Xian-Wen Yang

**Affiliations:** 1Key Laboratory of Marine Biogenetic Resources, Third Institute of Oceanography, Ministry of Natural Resources, 184 Daxue Road, Xiamen 361005, China; hezhihui@tio.org.cn (Z.-H.H.); xulin@tio.org.cn (L.X.); humanyi@tio.org.cn (M.-Y.H.); xiemingmin@tio.org.cn (M.-M.X.); haoyoujia888@163.com (Y.-J.H.); lishujin98@163.com (S.-J.L.); shaozongze@tio.org.cn (Z.-Z.S.); 2Yanjing Medical College, Capital Medical University, 4 Dadong Road, Beijing 101300, China; wujia@ccmu.edu.cn

**Keywords:** deep-sea, fungus, *Penicillium solitum*, anti-tumor, anti-food allergy

## Abstract

A systematic chemical investigation of the deep-sea-derived fungus *Penicillium solitum* MCCC 3A00215 resulted in the isolation of one novel polyketide (**1**), two new alkaloids (**2** and **3**), and 22 known (**4**–**2****5**) compounds. The structures of the new compounds were established mainly on the basis of exhaustive analysis of 1D and 2D NMR data. Viridicatol (**13**) displayed moderate anti-tumor activities against PANC-1, Hela, and A549 cells with IC_50_ values of around 20 μM. Moreover, **13** displayed potent in vitro anti-food allergic activity with an IC_50_ value of 13 μM, compared to that of 92 μM for the positive control, loratadine, while indole-3-acetic acid methyl ester (**9**) and penicopeptide A (**10**) showed moderate effects (IC_50_ = 50 and 58 μM, respectively).

## 1. Introduction

*Penicillium solitum* is a filamentous fungus associated with the decay of pomaceous fruits during storage [1]. As a matter of fact, it can infect fruit through wounds and cause significant economic losses [2]. Besides pome fruits such as apples and pears, this fungus was also isolated from other foods, including cheeses and processed meats [3,4]. Surprisingly, it can also be found under extremophilic circumstances: in the Berkeley Pit Lake (pH 2.7) [5] and the maritime Antarctic [6].

Chemical investigation of this fungus provided a broad spectrum of secondary metabolites, including compactin (known as mevastatin or ML-236B, which is utilized for the production of an important cholesterol-lowering drug, pravastatin) [7] and its analogues [8,9], in addition to sesquiterpenoids [5], alkaloids [10], and polyketides etc. [11].

*Penicillium solitum* MCCC 3A00215 is a deep-sea-derived fungus from the Northwest Atlantic Ocean (−3034 m). A previous study on this strain provided a unique 6/6/6/6/5-pentacyclic steroid [12]. In order to discover more novel compounds, a further chemical investigation was conducted. As a result, three new (**1**–**3**) and 22 known (**4**–**25**) compounds (Figure 1) were obtained. By comparison of the NMR and MS data with those published in the literature, the known compounds were determined to be (−)-solitumidines D (**4**) [10], ML-236A (**5**) [13], solitumidine A (**6**) [10], methyl-2-([2-(1H-indol-3-yl)-ethyl]carbamoyl)acetate (**7**) [14], solitumine A (**8**) [10], indole-3-acetic acid methyl ester (**9**) [15], penicopeptide A (**10**) [16], (2′*S*)-7-hydroxy-2-(2-hydroxypropyl)-5-methylchromone (**11**) [17], hydroxypropan-2′,3′-diol orsellinate (**12**) [18], viridicatol (**13**) [19], viridicatin (**14**) [19], (−)-cyclopenol (**15**) [20], cyclopenin (**16**) [21], 3-benzylidene-3,4-dihydro-4-methyl-1*H*-1,4-benzodiazepine-2,5-dione (**17**) [16], *β*-sitosterol-3-*O*-*β*-d-glucopyranoside (**18**) [22], cerebrosides C (**19**) [23], methyl-2,4-dihydroxy-3,5,6-trimethylbenzoate (**20**) [24], felinone A (**21**) [25], xylariphilone (**22**) [26], 5,6-dihydroxy-2,3,6-trimethylcyclohex-2-enone (**23**) [27], (*R*)-mevalonolactone (**24**) [28], and 3-methyl-2-penten-5-olide (**25**) [29]. Here, we report the isolation, structure, and bioactivities of these 25 compounds.

## 2. Results and Discussion

Compound **1** had a molecular formula C_19_H_30_O_5_, as established by its positive HRESIMS at *m/z* 361.1989 [M + Na]^+^, requiring five degrees of unsaturation. The ^1^H and ^13^C NMR spectroscopic data (Appendix A, Table 1) revealed the presence of one methyl doublet [*δ*_H_ 0.89 (d, *J* = 6.8 Hz, H_3_-16); *δ*_C_ 14.3 (q, C-16)], one methoxyl [*δ*_H_ 3.70 (s, OMe); *δ*_C_ 52.1 (q, OMe)], six *sp*^3^ methylenes, nine methines including three aliphatic [*δ*_H_ 1.77 (m, H-1), 2.19 (brd, *J* = 11.8 Hz, H-8a), 2.37 (m, H-2); *δ*_C_ 32.1 (d, C-2), 37.8 (d, C-1), 40.0 (d, C-8a)], three olefinic [*δ*_H_ 5.47 (brs, H-5), 5.69 (dd, *J* = 9.4, 6.1 Hz, H-3), 5.91 (d, *J* = 9.4 Hz, H-4); *δ*_C_ 124.5 (d, C-5), 129.9 (d, C-4), 133.6 (d, C-3)] and three oxygenated [*δ*_H_ 3.80 (m, H-11), 4.19 (m, H-13), 4.22 (m, H-8); *δ*_C_ 65.2 (d, C-8), 68.1 (d, C-13), 71.1 (d, C-11)] ones, and two non-protonated carbons with one olefinic [*δ*_C_ 135.1 (s, C-4a)] and one carbonyl [*δ*_C_ 173.9 (s, C-15)] group. Altogether, the ^1^H and ^13^C NMR spectra provided 19 carbons, categorized as one methyl, one methoxyl, six methylenes, nine methines, and two quaternary carbons.

In the COSY spectrum, correlations were observed for H-5/H_2_-6/H_2_-7/H-8/H-8a/H-1/H_2_-9/H_2_-10/H-11/H_2_-12/H-13/H_2_-14 and H-1/H-2/H_3_-16/H-3/H-4, which constructed a long chain of C-5/C-6/C-7/C-8/C-8a/C-1/C-9/C-10/C-11/C-12/C-13/C-14 and C-1 via C-2 to C-16/C-3/C-4 (Figure 2). The segment and the methoxyl moiety could be connected on the basis of the HMBC correlations of H-4 to C-8a/C-4a/C-5, H_2_-14 and 15-OMe to C-15 (Figure 2). Therefore, the planar structure of 1 was established as a methyl ester of acyclic form of ML-236A (5) [13], which was previously prepared in the lab by the saponification of ML-236A in 0.1 N NaOH at 50 °C for 2 h [30].

The relative configuration of **1** was supposed to be the same as that of **5,** according to the NOESY correlations of H-8a to H-8/H-9a/H_3_-16 and H_3_-16 to H_2_-9. On the basis of the similar optical rotation values of **1** (+62.7) and 5 (+73.3), and further by comparison of their electronic circular dichroism (ECD) spectrum (Figure 3), **1** was then established to be 15-*O*-methyl ML-236A.

Compound **2** was assigned the molecular formula C_2__0_H_2__7_N_3_O_5_ on the basis of the [M − H]^−^ ionic peak at *m/z* 388.2821 in its negative HRESIMS spectrum, suggesting nine degrees of unsaturation. The ^1^H and ^13^C NMR spectroscopic data, by the aide of the HSQC and ^1^H–^1^H COSY spectra, showed characteristics of a 1,2-disubsituted benzoic unit [*δ*_H_ 7.19 (t, *J* = 7.8 Hz, H-5), 7.60 (t, *J* = 7.8 Hz, H-6), 8.04 (t, *J* = 7.8 Hz, H-4), 8.54 (t, *J* = 7.8 Hz, H-7); *δ*_C_ 120.1 (d, C-7), 122.6 (s, C-7a), 122.8 (d, C-5), 131.4 (d, C-4), 134.5 (d, C-6), 139.9 (s, C-3a)], an isoprene [*δ*_H_ 1.32 (s × 2, C-11, 12), 5.25 (d, *J* = 17.4 Hz, H-10a), 5.29 (d, *J* = 10.6 Hz, H-10b),6.08 (d, *J* = 17.4, 10.6 Hz); *δ*_C_ 24.5 (q × 2, C-11, 12), 46.2 (s, C-8), 114.6 (t, C-10), 142.4 (d, C-9)], glutamic acid [*δ*_H_ 1.77−1.95 (m, H_2_-18), 2.23 (m, H_2_-17), 3.19 (m, H-19); *δ*_C_ 27.0 (t, C-18), 31.8 (t, C-17), 53.7 (d, C-19), 169.6 (s, C-20), 172.1 (s, C-16)], *β*-aminopropanone [*δ*_H_ 3.22 (t, *J* = 6.6 Hz, H_2_-13), 3.37 (m, H_2_-14), 8.16 (t, *J* = 4.9 Hz, H-15); *δ*_C_ 34.6 (t, C-14), 39.3 (t, C-13), 203.2 (s, C-3)], and one acylamide [*δ*_H_ 11.5 (s, H-1); *δ*_C_ 174.8 (s, C-2)]. These five fragments could be connected by the HMBC correlations of H_3_-11/H_3_-12 to C-2, H-1 to C-2/C-3a/C-7/C-7a, H_2_-4 to C-3, and H-14 to C-16 to construct the planar structure of **2** (Figure 4), the same as solitumidine D [10], namely **4**, which was simultaneously obtained along with **2** by HPLC using the A4-5 chiral column. Since the specific optical rotation of **2** was +6, opposite to that of **4** (−7) in the same concentration of MeOH (*c* 0.10), **2** was then deduced to be the enantiomer of **4**. Accordingly, **2** was determined as (+)-solitumidine D.

Compound **3** presented its molecular formula asC_21_H_28_N_2_O_4_ by the positive HRESIMS at *m/z* 395.1947 [M + Na]^+^. The ^1^H and ^13^C NMR spectra consisted of signals almost the same as those of solitumidine B [10] except for an additional methoxyl unit. This was confirmed by the HMBC correlation of 20-OMe (*δ*_H_ 3.73, s) to C-20 (*δ*_C_ 176.0, s). Furthermore, by extensive analysis of the COSY and HMBC NMR spectra (Figure 4), **3** was determined as 20-*O*-methyl solitumidine B. Since the optical rotation value for solitumidine B was −55 in MeOH, while it was 0 for **3** in the same solvent, **3** was supposed to be a racemic mixture. As such, it was subjected to further isolation by HPLC with chiral columns. Yet, **3** seemed to be inseparable as it exhibited only one peak using several different mobile phases in A3-5 and A4-5 chiral columns, the latter of which was utilized to successfully isolate **2** from its enantiomer, **4**. On the basis of the above evidence, **3** was then named as (±)-solitumidine E.

All isolates were tested for antiproliferative effect against 17 human tumor cell lines of A431, A549, MB231, MCF-7, PANC1, HepG2, HCT116, H460, H1299, QGY-7701, BGC823, SKGT4, A375, U2OS, HL-60, K562, and KYSE450 under the concentration of 20 μM and were tested for anti-food allergic activity under the concentration of 50 μM. Notably, viridicatol (**13**) showed significant cytotoxic activities against PANC1, Hela and A549 cells with IC_50_ values of 18, 19, and 24 μM, respectively. 

Moreover, compounds **1**–**25** were also tested in vitro for anti-food allergic activity. Indole-3-acetic acid methyl ester (**9**) and penicopeptide A (**10**) showed modest activity (IC_50_ = 50 and 58 μM, respectively), while **13** displayed a potent effect with an IC_50_ value of 13 μM, compared to that of 92 μM for loratadine, an anti-food allergic medicine in clinic. In fact, viridicatol isolated from another deep-sea-derived fungus, *Penicillium griseofulvum* MCCC 3A00225, showed a significant anti-food allergic effect in the RBL-2H3 cell model and the ovalbumin-induced food allergy mouse [31]. Therefore, it may represent a novel therapeutic for allergic diseases.

## 3. Materials and Methods

### 3.1. General Experimental Procedures

NMR spectra were recorded on a Bruker 400 MHz spectrometer. The HRESIMS spectra were recorded on a Waters Q-TOF mass spectrometer (Xevo G2). Optical rotations were obtained with an Anton Paar polarimeter (MCP100). ECD spectra were measured on a Chirascan spectrometer. The semi-preparative HPLC was conducted on an Agilent instrument (1260) with different kinds of columns (COSMOSIL 5 C18-MS-II, Nacalai Tesque, Japan; ColumnTek^TM^ Chiral A3-5 and A4-5). Column chromatography was performed on silica gel, Sephadex LH-20, and ODS.

### 3.2. Fungal Identification, Fermentation, and Extract

The fungus *Penicillium solitum* MCCC 3A00215 was isolated from a sediment sample of the Northwest Atlantic Ocean (−3034 m, W 44.9801°, N 14.7532°). For the large-scale fermentation procedure, see our recently published literature [12]. The crude extract (200 g) was subjected to column chromatography on silica gel using petroleum ether (PE), CH_2_Cl_2_, EtOAc to provide a CH_2_Cl_2_-soluble extract (11 g) and a EtOAc-soluble extract (114.5 g), respectively.

### 3.3. Isolation and Purification

The CH_2_Cl_2_ crude extract was separated into six fractions (Fr.A−Fr.F) via medium pressure liquid chromatography (MPLC, 460 mm × 36 mm) with gradient PE-EtOAc (5:1→1:5). Subfractions Fr.B-Fr.F were subsequently purified by column chromatography (CC) over Sephadex LH-20 (1.5 m × 3 cm; CH_2_Cl_2_-MeOH, 1:1) followed by semi-preparative HPLC with MeOH-H_2_O (40%→100%) to provide **10** (280 mg), **11** (2 mg), **17** (57 mg), **20** (3 mg), **21** (2 mg), **22** (1.5 mg), **23** (2 mg), and **25** (4.4 mg).

The EtOAc part was subjected to MPLC (460 mm × 46 mm) on silica gel with gradient CH_2_Cl_2_-MeOH (100%→50%) to obtain five fractions (Fr.1−Fr.5). Fraction Fr.1 (1 g) was separated by CC over Sephadex LH-20 (1.5 m × 3 cm, CH_2_Cl_2_-MeOH, 1:1) and subsequently purified by recrystallization to give **14** (60 mg). Fraction Fr.2 (4 g) was separated by CC over ODS (310 mm × 5 mm; MeOH-H_2_O, 10%→100%) and Sephadex LH-20 (1.5 m × 2 cm, MeOH), followed by semi-prep. HPLC (MeOH-H_2_O, 40%→100%) afforded **1** (22 mg), **3** (3 mg), **5** (6 mg), **7** (2 mg), **9** (12 mg), **12** (2 mg), **13** (1.4 g), **15** (3 mg), **16** (10 mg), **18** (3 mg), and **19** (50 mg), while **2** (4 mg), **4** (2 mg), **6** (30 mg), **8** (27 mg), and **24** (5 mg) were isolated from Fr.4 (8 g) by CC on ODS (310 mm × 5 mm; MeOH-H_2_O, 10%→100%) and Sephadex LH-20 (1.5 m × 2 cm, MeOH), followed by semi-prep. HPLC with chiral column A4-5 (MeOH-H_2_O, 40%→80%).

15-*O*-methyl ML-236A (**1**): colorless oil; [*α*]20D +62.7 (c 0.30, MeOH); UV (MeOH) *λ*max (logε) 237 (3.08) nm; CD (MeOH) (Δε) 204 (−1.90), 233 (+0.53), 236 (+0.53), 245 (+0.40) nm; ^1^H and ^13^C NMR data, see Table 1; HRESIMS *m/z* 361.1989 [M + Na]^+^ (calcd for C_19_H_30_O_5_Na, 361.1991).

(+)-solitumidine D (**2**): white amorphous solid; [*α*]20D +6 (*c* 0.10, MeOH); UV (MeOH) *λ*max (logε) 231 (4.73) nm, 261 (4.25) nm, 325 (3.85) nm; CD (MeOH) (Δε) 203 (+2.62) nm; ^1^H and ^13^C NMR data, see Table 1; HRESIMS *m/z* 388.1877 [M−H]^−^ (calcd for C_2__0_H_2__6_N_3_O_5_, 388.1872).

(±)-Solitumidine E (**3**): white amorphous power; [*α*]20D 0 (*c* 0.44, MeOH); UV (MeOH) *λ*max (logε) 222 (4.10) nm, 261 (3.70) nm, 291 (3.52) nm; CD (MeOH) (Δε) 203 (−0.48), 299 (−0.04) nm; ^1^H and ^13^C NMR data, see Table 1; HRESIMS *m/z* 395.1945 [M + Na]^+^ (calcd for C_21_H_28_N_2_O_4_Na, 395.1947).

### 3.4. ECD Calculation

Conformational analysis was performed by the Sybyl-X 2.0 using the MMFF94S force field as reported [32]. Predominant conformers were relocated and confirmed at the B3LYP/6-31G(d) level. The theoretical ECD spectra were calculated with the time-dependent density functional theory (TD-DFT) in acetonitrile. The ECD spectrum was obtained by averaging each conformer using the Boltzmann distribution theory.

### 3.5. Cell Proliferation Assay

Cytotoxic activities of all isolates were conducted on 17 human tumor cell lines of A431, A549, MB231, MCF-7, PANC1, HepG2, HCT116, H460, H1299, QGY-7701, BGC823, SKGT4, A375, U2OS, HL-60, K562, and KYSE450 by the MTT method [33]. Paclitaxel was used as a positive control, and DMSO was used as a negative control. Different cancer cells were incubated on 96-well cell plates and cultured for 24 h. Thereafter, the cells were treated with different concentrations of tested compounds and controls. After 48 h, MTT (20 μL) was added to incubate for another 4 h. The supernatant was discarded softly, and the deposited formazan formed in the cells was dissolved with DMSO (100 μL). The absorbencies were measured at 490 nm.

### 3.6. Anti-Allergic Bioassay

The in vitro anti-food allergic experiment was performed as previously reported [32]. In brief, rat basophilic leukemia 2H3 (RBL-2H3) cells were incubated with dinitrophenyl (DNP)–immunoglobulin E (IgE) overnight. Then, the IgE-sensitized RBL-2H3 cells were pretreated with tested compounds and stimulated with DNP–bovine serum albumin (BSA). The bioactivity was quantified by measuring the fluorescence intensity of the hydrolyzed substrate in a fluorometer. Loratadine, a commercially available antiallergic medicine, was used as a positive control.

## 4. Conclusions

One new compactin analogue (**1**) and two previously unreported alkaloids (**2** and **3**), together with 22 known compounds (**4**–**2****5**), were isolated from the deep-sea-derived *Penicillium solitum* MCCC 3A00215. Viridicatol (**13**) exhibited weak cytotoxic activities against PANC-1, Hela, and A549 cells with IC_50_ values of 18, 19, and 24, respectively, while it showed remarkable anti-food allergic activity, with an IC_50_ value of 13 μM.

## Figures and Tables

**Figure 1 marinedrugs-19-00580-f001:**
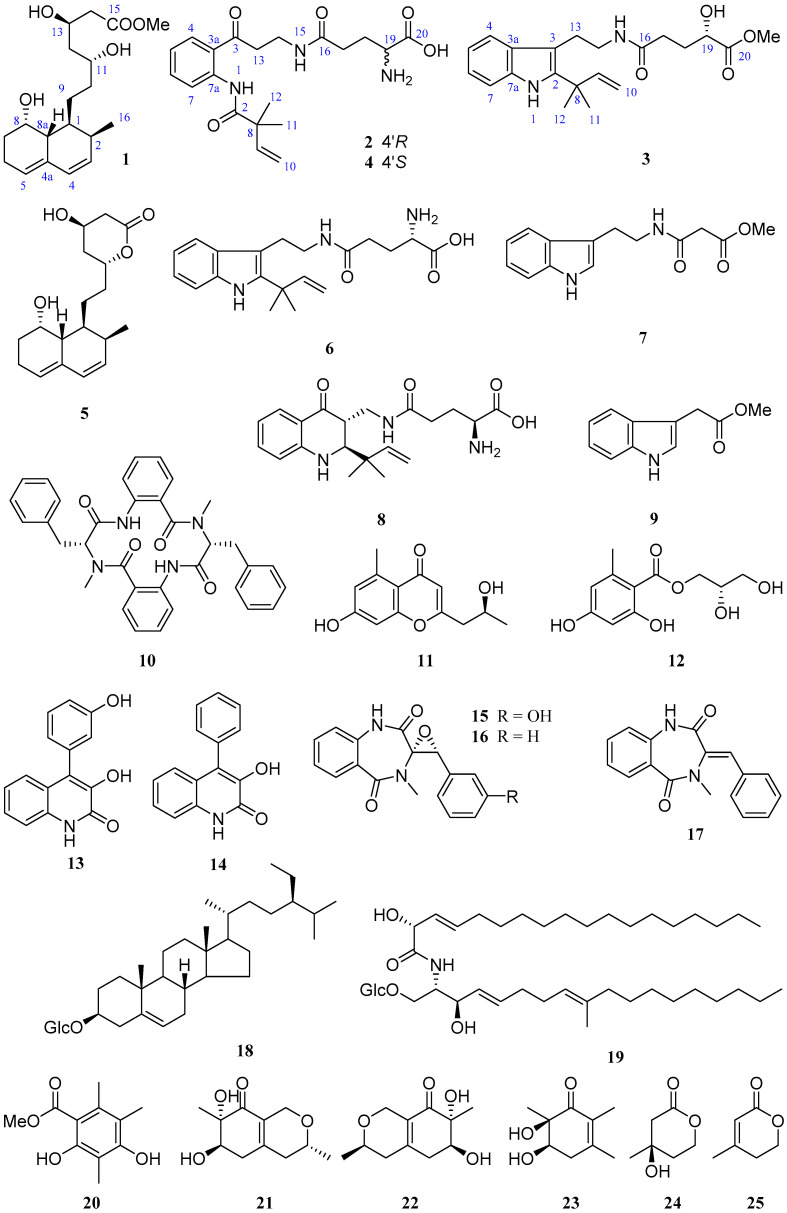
Compounds **1**–**25** from *Penicillium solitum* MCCC 3A00215.

**Figure 2 marinedrugs-19-00580-f002:**
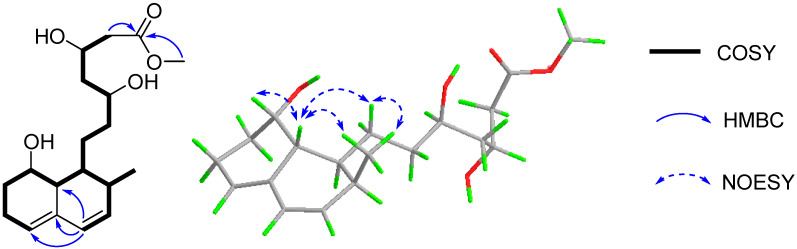
The key COSY, HMBC, and NOESY correlations of **1**.

**Figure 3 marinedrugs-19-00580-f003:**
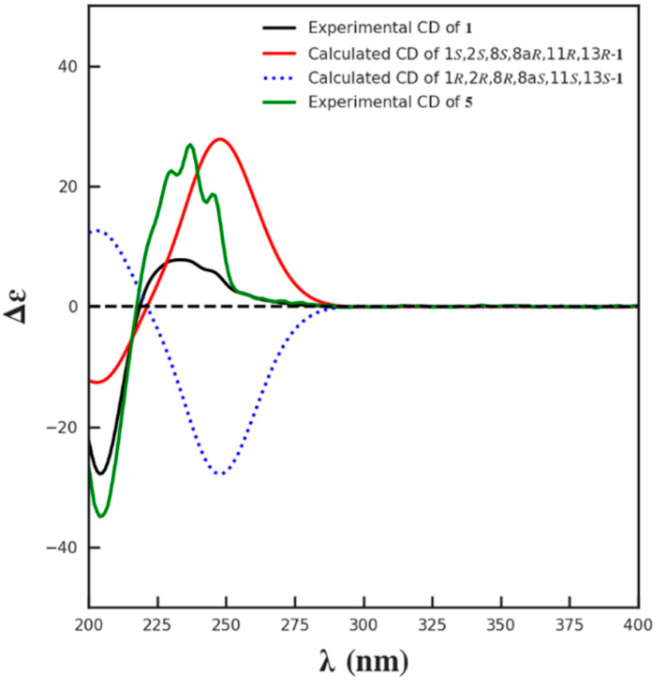
The calculated ECD spectrum of **1** and the experimental ECD spectra of **1** and **5**.

**Figure 4 marinedrugs-19-00580-f004:**
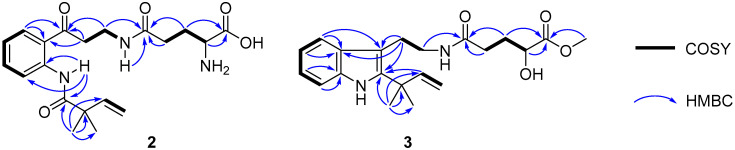
Key ^1^H–^1^H COSY and HMBC correlations of **2** and **3**.

**Table 1 marinedrugs-19-00580-t001:** ^1^H (400 Hz) and ^13^C (100 Hz) NMR data of **1**–**3** (*δ* in ppm, *J* in Hz within parentheses).

No.	1 *^a^*		2 *^b^*		3 *^a^*	
	δ_C_	δ_H_	δ_C_	δ_H_	*δ* _C_	*δ* _H_
1	37.8 CH	1.77 m		11.5 s		
2	32.1 CH	2.37 m	174.8 C		141.3 C	
3	133.6 CH	5.69 (dd, 9.4, 6.1)	203.2 C		108.4 C	
3a			139.9 C		130.8 C	
4	129.9 CH	5.91 (d, 9.4)	131.4 CH	8.04 (d, 7.8)	118.7 CH	7.49 (d, 7.9)
4a	135.1 C					
5	124.5 CH	5.47 (brs)	122.8 CH	7.19 (t, 7.8)	119.5 CH	6.95 (t, 7.9)
6	21.6 CH_2_	2.09 m, 2.33 m	134.5 CH	7.60 (t, 7.8)	121.7 CH	7.01 (t, 7.9)
7	30.6 CH_2_	1.68 m, 1.96 m	120.1 CH	8.54 (d, 7.8)	111.6 CH	7.28 (d, 7.9)
7a			122.6 C		136.4 C	
8	65.2 CH	4.22 m	46.2 C		40.1 C	
8a	40.0 CH	2.19 (brd, 11.8)				
9	25.0 CH_2_	1.33 m, 1.83 m	142.4 CH	6.08 (dd, 17.4, 10.6)	147.8 CH	6.18 (dd, 17.4, 10.6)
10	35.5 CH_2_	1.41 m; 1.54 m	114.6 CH_2_	5.25 (d, 17.4);	111.7 CH_2_	5.06 (dd, 17.4, 1.5);
				5.29 (d, 10.6)		5.09 (dd, 10.6, 1.5)
11	71.1 CH	3.80 m	24.5 CH_3_	1.32 s	28.5 CH_3_	1.54 s
12	44.8 CH_2_	1.64 m	24.5 CH_3_	1.32 s	28.5 CH_3_	1.54 s
13	68.1 CH	4.19 m	39.3 CH_2_	3.22 (t, 6.6)	26.3 CH_2_	2.99 (dd, 8.2, 7.6)
14	43.1 CH_2_	2.46 m, 2.57 m	34.6 CH_2_	3.37 m	41.6 CH_2_	3.36 (dd, 9.9, 7.6)
15	173.9 C			8.16 (t, 4.9)		
16	14.3 CH_3_	0.89 (d, 6.8)	172.1 C		175.1 C	
17			31.8 CH_2_	2.23 m	32.6 CH_2_	2.27 (ddd, 11.0, 8.5, 8.0)
						2.29 (ddd, 11.0, 6.4, 4.4)
18			27.0 CH_2_	1.77−1.95 m	31.2 CH_2_	1.88 (ddt, 14.3, 8.5, 6.4)
						2.06 (ddt, 14.3, 8.0, 4.4)
19			53.7 CH	3.19 m	71.0 CH	4.16 (dd, 8.0, 4.4)
20			169.6 C		176.0 C	
OMe	52.1 CH_3_	3.70 s			52.5 CH_3_	3.73 s

*^a^* Recorded in CD_3_OD. *^b^* Recorded in DMSO-*d*_6._

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
