# Peer review of "Chemical Constituents of the Deep-Sea-Derived Penicillium solitum"

_marinedrugs, 2021, doi:10.3390/md19100580_

Round 1

Reviewer 1 Report

THe paper describes a systematic chemical investigation on the deep-sea-derived fungus Penicillium solitum and the identification  of one novel compactin-type polyketide (1), two new alkaloids (2 and 3) and 22 known (4–25) compounds. 

The structure determination of the compounds is clear and the manuscript is acceptable for publication after addressing the following minor requests:

  1. The text similarity (plagiarism) in the text is too high. The paper should be rewritten thoroughly and the text similarity should be minimal. A copy of text similarity report is attached here
  2.  Abstract "investigation on the" should be "investigation of the"
  3. Introduction: The references for the known compounds (4-25) should be cited after the first mention of the compounds in the second line from the bottom in the first page.
  4. The configuration of the OH group in compound 3 should be determined. the compound did not possesses any optical activity (Optical rotation value is 0), which means that the compound in optically inactive (i.e. racemic mixture). If nit the authors should address this point and assign the configuration of the OH group.
  5. Page 3. Line 3 from the bottom 14-OMe is wrong. It should be OMe only. The methoxy group is not attached to C-14.
  6. Page 5; line 5 and 6: the number of 5'-OMe and C-5' and 5'-O-methyl is wrong. There is no "dash " in the drawn structure of compound 3. Please correct.
  7. All 1H and 13C NMR spectra of the know compounds (4-22) should be included in the supporting information.

Reviewer 2 Report

The aim of this work was to isolate, identify and test some bioactivities of compounds isolated from an unusual source, the deep-sea-derived fungus Penicillium solitum. Three of the 25 compounds were novel. They were tested for activity against 17 tumor cell lines in vitro. Comparison with potency against normal cells such as lymphocytes of fibroblasts would have been useful. The 24-hr treatment time only allows for about 1 doubling time and therefore might have underestimated these potencies using the metabolic dye MTT. The IC50 for paclitaxel was not given but one would expect nM value if the cells were exposed for 6-7 days.  Potency of 15-20 uM in vitro is not great for anticancer potential. In the absence of finding potent bioactivities, the overall significance of the work is low.

Apart from these limitations the experimental approach was appropriate, thorough and extensive, using appropriate controls. The impact of the work could be increased by the following.

  1. Because of its close similarity to compactin (mevastatin) as well as isolation from a Penicillium the ability of compound 1 to inhibit HMG-CoA reductase should be tested.
  2. The significance of the in vitro anti-food allergic activity of Compound 13 vs larotidine merits further discussion about its potential as a therapeutic.
  3. Antibacterial and/or antibacterial activities are typically found in Penicillium and such assays could be applied at least to the novel compounds.

Round 2

Reviewer 2 Report

No further comment